# A petascale automated imaging pipeline for mapping neuronal circuits with high-throughput transmission electron microscopy

Wenjing Yin [1,4✉], Derrick Brittain[1,4], Jay Borseth[1,4], Marie E. Scott[1], Derric Williams[1], Jedediah Perkins[1], Christopher S. Own[2], Matthew Murfitt[2], Russel M. Torres [1], Daniel Kapner[1], Gayathri Mahalingam[1], Adam Bleckert[1], Daniel Castelli[1], David Reid[1], Wei-Chung Allen Lee [3], Brett J. Graham[3], Marc Takeno [1], Daniel J. Bumbarger[1], Colin Farrell[1], R. Clay Reid [1✉] & Nuno Macarico da Costa [1✉]

Electron microscopy (EM) is widely used for studying cellular structure and network connectivity in the brain. We have built a parallel imaging pipeline using transmission electron microscopes that scales this technology, implements 24/7 continuous autonomous imaging, and enables the acquisition of petascale datasets. The suitability of this architecture for large-scale imaging was demonstrated by acquiring a volume of more than $1\,mm^3$ of mouse neocortex, spanning four different visual areas at synaptic resolution, in less than 6 months. Over 26,500 ultrathin tissue sections from the same block were imaged, yielding a dataset of more than 2 petabytes. The combined burst acquisition rate of the pipeline is 3 Gpixel per sec and the net rate is 600 Mpixel per sec with six microscopes running in parallel. This work demonstrates the feasibility of acquiring EM datasets at the scale of cortical microcircuits in multiple brain regions and species.

[1] Allen Institute, Seattle, WA, USA. [2] Voxa, Seattle, WA, USA. [3] Harvard Medical School, Boston, MA, USA. [4] These authors contributed equally: Wenjing Yin, Derrick Brittain, Jay Borseth. ✉email: wenjingy@alleninstitute.org; clayr@alleninstitute.org; nunod@alleninstitute.org

Serial-section electron microscopy has a long history of elucidating brain structure and connectivity[1-3]. Sixty years after it was first used in neuroscience[3], it remains the gold standard method for describing neuronal morphology and identifying synapses. With recent technical advances, electron microscopy has been through a renaissance. The first complete connectome, of an entire worm nervous system[2], has been updated and extended[4,5], while progress in system automation[6-11] has enabled large scale studies of the nervous systems in the fly, fish, bird, and mammal[7,10,12-25]. In some cases, such as for the fruit fly[10], the datasets include the totality of the adult brain, while in mammals the largest dataset ($\sim 0.07$ mm$^3$) includes a fraction of a thalamic region[23] (incidentally the same structure, the dorsal lateral geniculate nucleus, was the target of one of the largest 20th century reconstructions in the mammalian brain[26]).

A longstanding goal has been to reconstruct a complete local cortical microcircuit. Depending on the species and cortical region, this might require imaging one cubic millimeter or more, at least an order of magnitude larger than previous datasets. Imaging at the scale of a cubic millimeter is best done on EM systems capable of handling thousands of sections at a time, offering fast imaging without sacrificing resolution. Such EM systems should also provide high degrees of automation and reliability in order to permit unsupervised continuous operation.

In the past two decades, high-throughput EM has been transformed by powerful developments in technology. For example, serial block-face scanning electron microscopy (SB-SEM), with the ultramicrotome in the microscope[6,27], has been used for multiple regions and species[19,21,28]. The use of SEM back-scattered electrons at low electron energies confines the depth of the electron signals at the surface of the block and yields high z-resolution[19]. An alternative block-face approach uses focus ion beam scanning electron microcopy (FIB-SEM) and generates images with superior z-axis resolution and isotropic voxels, which accelerates the downstream 3D reconstruction and proofreading[8,17]. Although SB-SEM and FIB-SEM are successful approaches for imaging small-volume datasets, they still face challenges in expanding to large scales such as the cubic millimeter volume discussed in this work, and FIB-SEM requires highly customized expensive facilities and engineering to maintain long-term system stability. One risk of block-face approaches is that they are destructive methods, unable to reimage sections if imaging errors occur or for further study. This is in contrast with serial-section approaches, using either scanning or transmission electron microscopy (TEM)[7,10,11,22,23], which provide the opportunities for reimaging. For SEM, the highest throughput has been achieved with multi-beam instruments[29,30], which were originally designed for semiconductor lithography, reverse engineering, and wafer defect inspection, but recently found great potential in neuronal circuit reconstruction. The multi-beam SEMs have achieved burst imaging rates at 0.45 Gpixel per sec in mouse brain[29,30], showing a remarkable advancement in acquisition speed and capability of imaging large sample areas. There are however factors that lead to long-term effective continuous acquisition rates still yet to be demonstrated, such as stage motion accuracy and settling time, focus variation across montage, system stability for continuous 24/7 operation, and maintenance overhead. The alternative to scanning is TEM. TEM is inherently parallel while SEM acquires data serially, one pixel at a time[27,31]. Most standard TEMs can handle 1–5 sections per load, but recently serial-section TEM has been designed for handling thousands of sections[10,32,33], thereby allowing continuous imaging.

Our overarching strategy is to automate high-resolution TEMs with precise sample handling and fast cameras to increase the imaging throughput and quality. TEMs achieve perhaps the highest signal-to-noise ratio[10], especially for fast imaging, but commercially available TEMs are neither designed nor optimized

to efficiently image serial sections at a large scale. To achieve this we built upon the original TEMCA[7] design to create a system that has full automation, simple modular design, and systems-level feedback for error correction and quality control, allowing the scope to operate with little user intervention after the initial experimental setup.

We have implemented a distributed platform of multiple high-throughput automated Transmission Electron Microscopes (auto-TEMs) that can simultaneously image multiple sections from the same block of tissue. We refer to this pipeline as Parallel Imaging using Transmission Electron Automated Microscopy (piTEAM). The pipeline offers an exceptional combination of high speed, high resolution, low signal-to-noise, and cost efficiency to map the brain structure and neural circuits. Throughput can be increased either by upgrading individual components, such as the camera of each microscope, or by scaling horizontally with more microscopes. Parallel imaging is coordinated over multiple scopes, adding robustness to system failure and downtime, since no single microscope is a bottleneck for data production. The piTEAM infrastructure integrates imaging automation at the level of TEM control, calibration, system monitoring, and the databases of samples and 2D montages. In the current platform, the burst imaging rate of each microscope is 0.5 Gpixel per sec (image acquisition only) and the net continuous imaging rate is 0.1 Gpixel per sec (including image acquisition and operational overhead time for image correction, stage movement, image capture, quality control, and post processing). This imaging pipeline has proved its reliability by imaging multiple datasets with regions of interest (ROIs) of more than 1 mm$^2$ per section.

## Results

**Development of an automated transmission electron microscope.** The imaging platform described here uses a standard JEOL 1200EXII 120 kV TEM that has been modified with customized hardware and software. The key hardware modifications are: (1) an extended column and custom electron-sensitive scintillator that produce a tenfold increase in the field-of-view with negligible impact on spatial resolution; (2) a single, large-format CMOS camera outfitted with a low distortion lens that reduces image acquisition time to 50–150 ms; (3) a nano-positioning sample stage that offers fast, high-fidelity montaging of large tissue sections; and (4) an advanced reel-to-reel tape translation system that accurately locates each section using index barcodes for random access on the GridTape[32].

*Optics.* The column extension of an autoTEM is similar to the original TEMCA design of Bock et al.[7] but with an even larger custom 30.5 cm scintillator that allows over 20 microns (at 4 nm per pixel resolution) on each side to be captured by a single CMOS camera. The camera is positioned using a stage with precision of micrometers for linear and rotational adjustment, which allows for efficient light gathering using optical lenses with large apertures and consequent shallow depth of field. Field distortion is minimized using a high-quality commercial lens (Zeiss Otus, 55 mm f/1.4). We chose a single large sensor rather than the original $2 \times 2$ camera array[7] to eliminate the computational overhead required to queue, process, and quality-control images from separate cameras. For the mm$^3$ mouse dataset described below, we initially used 20 Mpixel cameras (XIMEA, CMOSIS CMV20000), and later upgraded to 50 Mpixel cameras (CMOSIS CMV50000) that immediately doubled throughput (burst rate: $\sim 0.5$ Gpixel per sec, net rate 0.1 Gpixel per sec per microscope). More than 70 montages, each 1 mm$^2$, could be imaged per day per scope. These large-frame high-speed sensors were unavailable when the first TEMCA prototype was created[7] and demonstrate how the parallel imaging approach of TEM

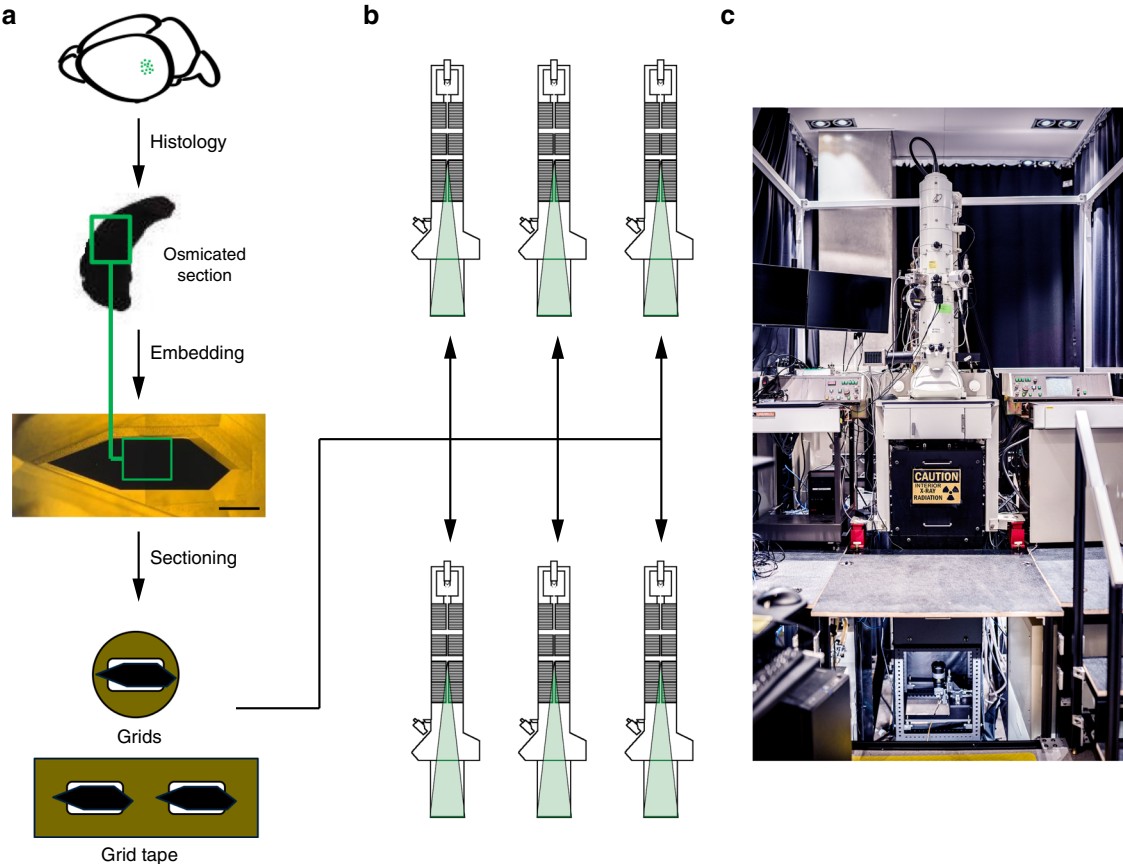

**Fig. 1 Experimental pipeline from sample preparation to imaging. a** Sample preparation. A mouse brain with the region of interest (ROI) indicated with green dots is sliced into thick brain sections (thickness ≥1 mm). The thick brain sections undergo histology with osmium protocol and become dark (the green square represents the ROI to be imaged with the electron microscope). After embedding, the block of tissue (1 mm scale bar) is trimmed in a hexagonal shape and prepared for ultrathin sectioning onto grids or GridTape. **b** Schematic cross-section of six distributed autoTEMs. **c** Photograph of an autoTEM system in the EM suite.

leverages the tremendous progress in camera sensor technologies. The use of off-the-shelf cameras decreases cost while permitting upgrades as new sensors arrive on the market.

*Sample handling.* We have implemented two different stage designs to achieve high sample packing density and automation: (a) a piezo two-axis stage with high-density standard sample grids (Voxa GridStage Sprite™, more details discussed in Methods) and (b) a TEM tape-compatible stage (Voxa GridStage™ Reel) that can handle thousands of sections in one sample load. The 1 mm³ data collection detailed below adopted a GridTape approach: Automated Tape Collecting Ultramicrotome (ATUM)[11] for section collection[20,23]. This was pioneered by K. Hayworth et al.[34], and was newly engineered to be GridTape[32] developed by Lee et al.

For the reel-to-reel system (Fig. 1c and Supplementary Fig. 1b), we loaded 5500 sections per autoTEM in a single pump down cycle. The elimination of repeated sample loading allows for 24/7 continuous imaging over large sample sets. The tape automatically translates between two reels mounted at the opposite sides of the TEM column and is adjusted through a pinch drive motor system with speed tuning. Each section on the GridTape is uniquely identified by a barcode ID and moves through the channel of in-column GridStage for serial montage. GridStage encapsulates an optical barcode reader that scans through the section barcode ID. A customized search algorithm has been developed to enable bi-directional, sequential, and random searches of sections during dynamic tape translation to the desired position. In addition, mechanical clamps within the

GridStage mitigate micro-vibrations and eliminate tape slippage during fast stage movement that can disrupt absolute position accuracy. The ON and OFF states of the clamp precisely synchronize with section montage and section translation to prevent imaging error. A customized tension sensor within the vacuum load lock provides dynamic monitoring on tape status and triggers section barcode reading: tape is tensioned during movement and is slack during montaging, which mechanically and thermally isolates the imaging region to facilitate high quality acquisition.

**A software infrastructure for petascale imaging via piTEAM.** An automated electron microscopy pipeline such as piTEAM requires data-driven, systems level control similar in principle to the fly-by-wire approach to automation in avionics using a closed feedback loop. A similar principle is used by our autoTEM system to control the state of the microscope without human intervention and dynamically change parameters to ensure consistent data quality and throughput. Live measurements are collected (e.g., focus score, pixel intensity histogram spread, brightness uniformity, beam centering, lens distortion within FOV) and the generated data are used to adjust microscope parameters to stay within given limits defined prior to imaging.

The piTEAM software infrastructure (Fig. 2a) has six core components that work in synchrony to provide continuous uninterrupted acquisition: (1) an image acquisition system (*pyTEM*); (2) real time image processing (*TEM graph*); (3) a graphical user interface (GUI); (4) databases; (5) facility and

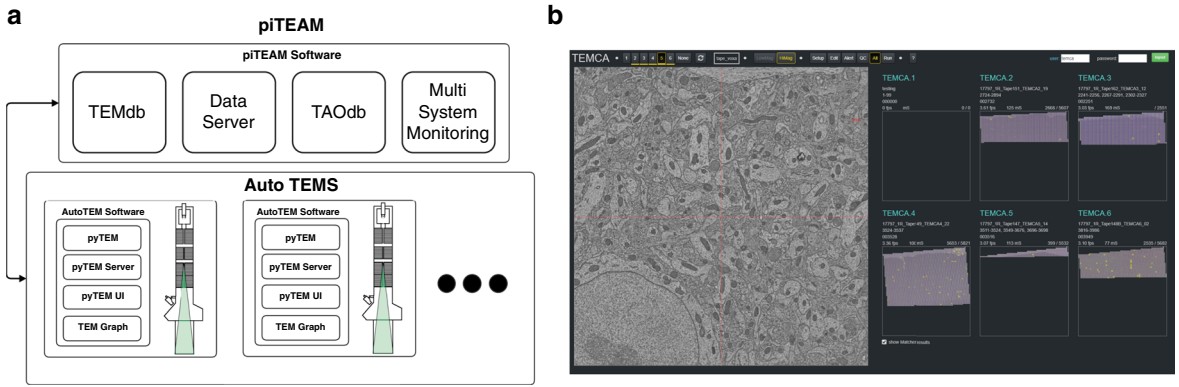

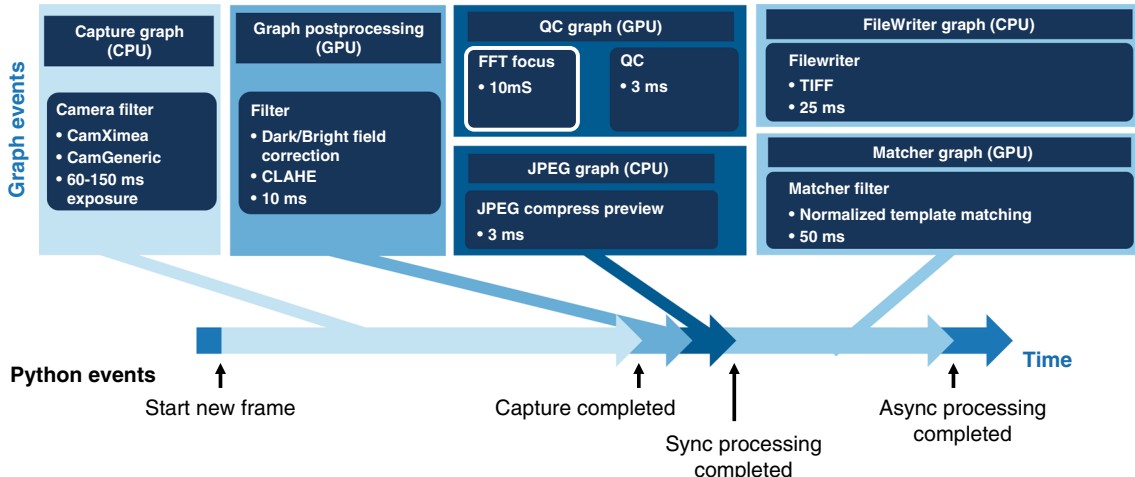

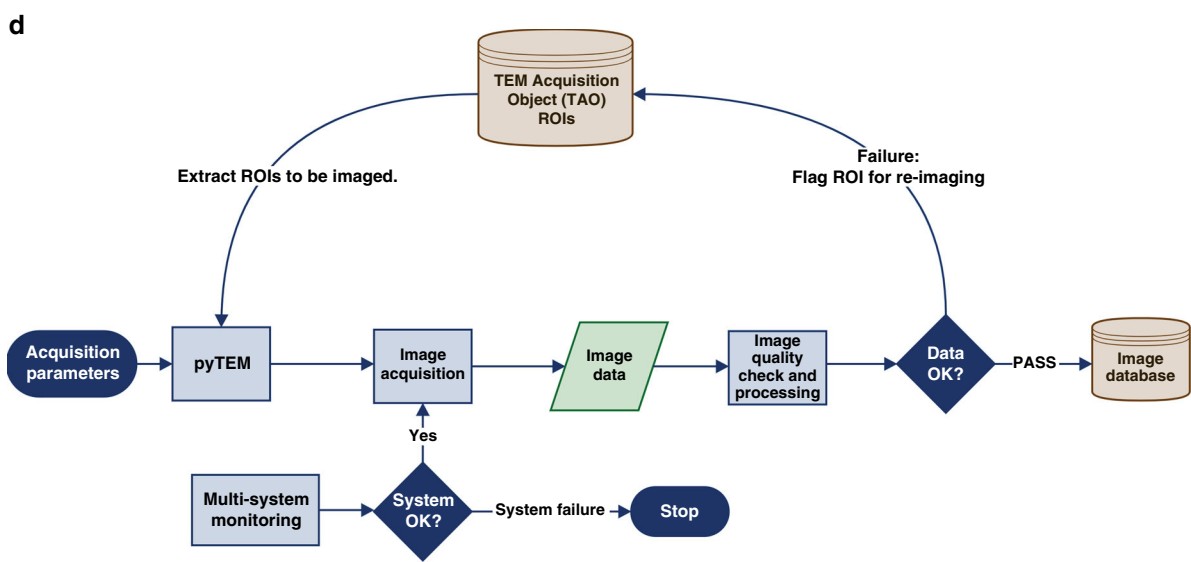

environment monitoring; and (6) a remote server (*pyTEM Server*). Below we provide a brief description of these core components with more details in the Methods section.

The imaging program pyTEM orchestrates the automated acquisition. pyTEM controls the microscope via serial ports, calculates the positions of the images to be captured, issues the commands to the stage and reel-to-reel systems, writes metadata files, sends status messages to monitoring system, and publishes preview images and status to the GUI. During system setup it includes functions to calibrate beam rotation, pixel size, and autofocus. The core of the image acquisition pipeline is TEM Graph (Fig. 2c). It allows the microscopes to perform several distinct image processing tasks in parallel, such as flat field correction, focus measurement, image statistics, template matching, lens distortion correction, and file writing. To control the image acquisition pipeline, we use a browser-based GUI (Fig. 2b

**Fig. 2 Imaging pipeline and workflow. a** The architecture of piTEAM pipeline. It is composed of distributed autoTEMs for parallel imaging, an image record database, data servers, a sample database (TAO) and Multi-System Monitor (MSM). On each individual autoTEM, imaging is operated through pyTEM (acquisition software), pyTEM server, pyTEM GUI and TEM Graph. **b** pyTEM GUI. The left EM image is a preview while the right is an example of parallel imaging on five systems for 1 mm² montage. The pyTEM GUI provides the user with an intuitive, web-based interface to perform manual imaging surveys as well as long serial montage runs containing hundreds or thousands of ROIs. From the web UI, any running autoTEM system can be observed and controlled. **c** TEM Graph key components. Images are acquired and loaded into GPU memory. A series of filter graphs apply corrections to the image (flatfield, down sampling for GUI preview). Separate graphs check image quality and statistics while the image is written to disk in parallel. **d** Closed-loop imaging workflow. After pyTEM receives ROIs and acquisition parameters, image acquisition is triggered, and image data are then analyzed on-the-fly on the acquisition computer. Rejected montages (those failing to meet QC thresholds) are flagged as a montage database instance to be re-imaged. If a montage passes inspection, it is sent to a data center for post-processing, alignment, and storage.

and Supplementary Figs. 2 and 3). It allows basic user guided remote operation of a microscope but also instructs the system to automate serial-section acquisitions. The GUI also displays quality control results and metadata about the current acquisition. The databases are crucial when handling datasets with tens to hundreds of millions of images, and the imaging pipeline relies on two databases to keep track of imaging, both of which can be accessed from GUI. First, the *TAO* (TEM Acquisition Object) database stores relevant information for each region of interest to be acquired. Second, *TEM* database holds records of ROIs that are acquired. The records include an overview of the tissue ROI, metadata about the acquisition and a flag to set the state of the quality of the acquisition. A user can review acquired sections, and either pass them to be ingested into the alignment pipeline or reject them for later re-imaging.

In order to sustain continuous and high quality imaging, it is also important to keep track of the stability of the microscope facility and environment, and therefore we integrated a multi-system monitor (MSM) (Supplementary Fig. 5) with the microscopes to ensure that the systems are operating within required environmental and mechanical bounds. A series of sensors that are physically installed on the microscope record data that are sent to a time series database with a graphical frontend. Alarms can be triggered if values pass set thresholds, which can command the pyTEM Server to stop acquisition if necessary. Finally, each running microscope has a corresponding instance of a server called the pyTEM Server. The server facilitates the communication between the GUI, remote monitoring system, database, and the pyTEM acquisition software. In addition, it republishes the images and the status streaming from pyTEM to an arbitrary number of clients and allows the control of TEMs by multiple users remotely.

**Fully automated closed loop imaging workflow and real-time QC.** Beyond the main task of automating TEMs, a further requirement for the imaging pipeline is to ensure that the multiple, concurrently active systems are consistently producing high quality image data and preserving sample integrity. Therefore, the piTEAM pipeline is designed with fully automated closed-loop feedback (Fig. 2d).

The digitally controlled microscopes first receive acquisition parameter inputs and a list of sample barcodes along with the ROI to image. Before each montage starts, the MSM assesses the microscope conditions and the system decides whether to proceed with imaging or to stop the acquisition and put the microscope in a safe condition. For each section, montage initialization steps are automatically carried out within minutes, including centroid finding, flatfield correction, and autofocus, the values of which are verified and self-corrected before montaging. For example, if the beam is displaced away from the center of the FOV due to thermal drift or filament aging, an automated beam centering routine repositions the beam. Deficient brightness or poor image focus trigger alerts on the montage status feedback

and the system automatically reattempts new brightfield or autofocus before montaging.

During montaging, the system will continuously loop through the following steps (Supplementary Fig. 4): (1) advancing the tape, (2) seeking a barcode and retrieving the corresponding TAO, (3) extracting the ROI, and (4) raster scan montaging ROI or triggering an abort when detecting an error state. Supplementary Movie 1 shows a live recording of five autoTEM systems running and collecting 1 mm² montages, reaching a fast acquisition rate of about four frames per second on average. Each acquired image is displayed in pyTEM GUI along with real time QC statistics such as focus score, tile overlap errors, etc. The QC information is written into a metadata file along with raw images and analyzed automatically after a montage completes. The pipeline makes a risk assessment on whether to proceed to the next indexed section based on user-defined screening criteria. If the QC result is good, the image data are sent on to the image database, otherwise the montage is marked as a QC failure. If the number of QC failures exceed a user-set threshold, the acquisition automatically stops for engineering review.

This automated imaging workflow was tested during the collection of tens of thousands of 2D montages over millions of individual tiles. The collection of such vast datasets allowed us to encounter various imaging errors and continuously improve image data QC metrics to catch them and apply fixes in time. Detailed imaging error examples are discussed in Supplementary Fig. 9. Such errors are best caught and corrected during imaging rather than during post-processing and data analysis, which may happen weeks or months after acquisition. Therefore, to ascertain whether the data generated are of sufficient quality for post-processing, a real-time QC software package has been developed.

***Real-time QC*** provides image quality metrics and system errors during data acquisition. The results are displayed on the pyTEM GUI and saved in the metadata file for each individual montage. To verify sufficient overlap between neighboring image tiles for 2D stitching, the GPU-based matcher filter utilizes normalized cross correlation template matching between tile edges along both X and Y directions (Fig. 3a). Real-time QC allows immediate visualization and identification of imaging problems at no additional cost to acquisition speed, because the typical template matching calculation overlaps with the next stage move.

Maps of the section and associated QC results are generated at the end of each montage. Figure 3b shows a typical visualization of the good vs. bad real-time template matching results. It is important to note that the matching succeeds even in areas without tissue such as on Luxel substrates.

Figure 3c–f show the QC outputs of matching overlap quality, image focus, and image offset in both *x* and *y* dimensions from a good 2D montage. The uniform color pattern across the frame for match quality (Fig. 3c) and match distance (Fig. 3e, f) indicate good overlap area in-between neighboring tiles. Figure 3d shows a fairly uniform focus map, with only tissue structures such as large

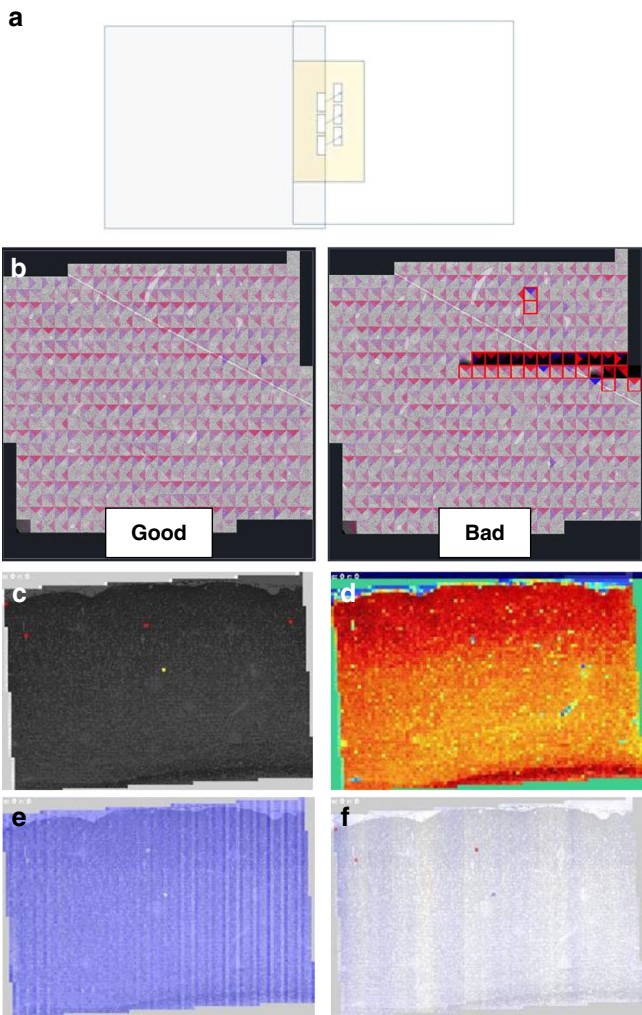

**Fig. 3 Real-time quality control (QC) for capturing image and system errors.** It utilizes the template matching to detect any tile overlap issue during acquisition, and FFT score to measure the focus quality of each tile. **a** Diagram representing the overlap region and template matching between two image tiles (blue boxes). The template search area (yellow box) is a region of twice the tile overlap (~13% for 20Mpix camera and 9% for 50Mpix). Three templates are used per edge, and the mean and standard deviation of the three matching vectors are returned from the filter. **b** Good vs. Bad real-time matcher results displayed on GUI: each triangle represents a matching result (top and down or left and right). Blue hues indicate that the template is found beyond the expected position, and red hues indicate that the template is found before the expected position, and the intensity represents the magnitude of the offset between the ideal and actual locations. To test the matching operation, we introduced two types of errors shown in the bad montage. The stage position was artificially perturbed on row 4, and then on rows 10–11 the beam was blocked, resulting in black tiles. If the number of matched templates or the standard deviation of the template match vectors fail to meet thresholds, the tile is marked as an error. Imaging problems are usually detected as flagged tiles in the quality map or non-uniform QC output maps. See example QC output maps: **c** matcher quality map; **d** focus map (color code is arbitrary as color is used only to identify any non-uniform pattern); **e** x-offset from ideal; **f** y-offset from ideal. Blue hue indicates positive stage offset comparing to the target position, and red hue indicates negative offset. The intensity represents the magnitude of the offset.

blood vessels being highlighted. For comparison, Supplementary Fig. 9 discusses a few QC failure examples. Overall, this real-time QC module enables an early identification of imaging errors.

The QC module described above minimizes errors and interruptions during imaging and ensures high quality and consistency of images. This has important consequences for the throughput of the image processing that happens after imaging, especially on such large datasets where tens to hundreds of millions of images have to be stitched into sections and aligned into a coherent 3D volume. For the datasets described in this manuscript we were able to compute more than 200 point correspondences along each edge of overlapping image tiles at a scale factor of 0.35, demonstrating high quality EM images are sufficient for the feature extractor to find good enough matching features in the overlapping region at a much lower resolution. It is also worth noting that an average stitching residual of 2.56 pixels was achieved on the cubic mm dataset, showing very good montage quality. In addition, the rough alignment for aligning the sections in 3D $z$-space was performed on montages that were scaled down to a resolution of just 1% of the actual resolution. Overall, superior quality of the images from the EM imaging pipeline allows the post-imaging processing to be performed much faster and cheaper computationally.

**Petascale data acquisition using piTEAM.** The first dataset collected by the piTEAM pipeline defined above contained 2500 tissue sections (250 μm × 140 μm, 40 nm thickness) that were imaged at 3.58 nm × 3.58 nm pixels. A single autoTEM system imaged the sections in 1 week with a 20 MP XIMEA camera and a GridStage Sprite sample holder (Supplementary Fig. 1 and Movie 2). The pilot dataset contained about 500 neurons and 3.2 million synapses, spanning over layer 2/3 mouse primary visual cortex, which have been analyzed to examine network connectivity of pyramidal and chandelier cells[15,16].

The biggest challenge for scaling from this 0.003 mm³ to 1 mm³ was throughput. We expanded the imaging platform from a single microscope to a multi-scope pipeline, which allowed us to scale the imaging system while maintaining the large electron flux required for fast exposure. We also transitioned from a low capacity (10 s of sections) stick-type GridStage Sprite[35,36] to a high capacity (1000 s of sections) GridStage reel-to-reel system for translating GridTape[32] between sections and then montaging. Over 26,500 sections were divided onto seven reels that were continuously imaged across five microscopes for almost 6 months. Using the 20 MP camera, the net image acquisition speed (including imaging, stage step-and-settle time, imaging overhead and image correction) averaged 3.3 frames per second (fps). Each frame contained 3840 × 3840 pixels, while each 1 mm² montage was composed of more than 5000 15 μm × 15 μm tiles (Fig. 4e) with an overlap of 13% between tiles in both $X$ and $Y$ directions. The 13% was chosen in order to ensure that stage imprecisions will still ensure a 7% overlap required in order to provide enough feature matches for stitching. The total file size of a single montage was about 80 GB producing a daily throughput of 3.6 TB per system for continuous imaging.

During production, three autoTEMs were upgraded with 50 MP camera sensors, which increased the frame size to 5408 × 5408 pixels (21 μm × 21 μm). Figure 4b–d show a high-res 50 MP tile from the dataset with neuronal somata and glia highlighted and zoomed-in areas at synapse level. The total number of tiles required per montage was reduced to ~2600 from more than 5000 at an overlap of 9% in both $X$ and $Y$, which maintained the same frame rate during montaging (Fig. 4f). The light-weight piezo stage not only preserves fast speed but also takes much shorter step-and-settle time at an average around 100 ms. The distribution of one stage

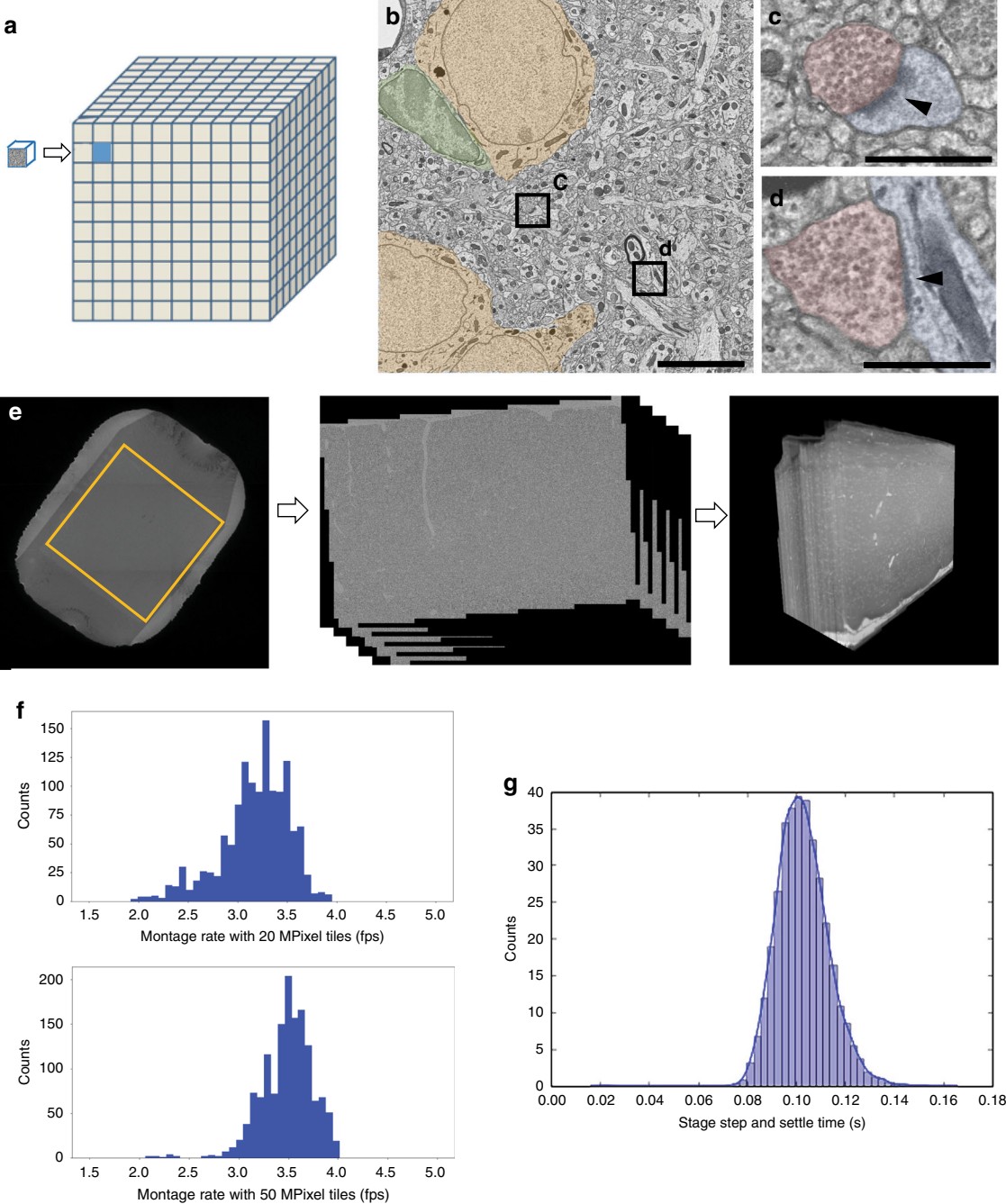

**Fig. 4 Imaging of a cubic millimeter of mouse cortex with piTEAM. a** Scaling from 0.003 mm³ to 1 mm³, a 300-fold volume increase; **b** High-resolution electron microscopy image tile from the 1 mm³ dataset with neuronal somata highlighted in yellow and glia in green (scale bar 5 μm). **c** Zoomed-in area from (**b**) showing synapse with dendritic spine (scale bar 1 μm); **d** Zoomed-in area from (**b**) showing synapse with dendritic shaft (scale bar 1 μm); **e** Low-mag EM image of an aperture with an ROI highlighted; 2D stacked montage minimap and aligned 3D volume. **f** Distribution of montage acquisition rate (frames per second) achieved during 1 mm³ production. The plots represent a sample size of over 1000 sections imaged by 20 Mpixel and 50 Mpixel cameras each. **g** Example of stage step-and-settle time distribution for GridStage.

example is shown in Fig. 4g. Table 1 is a comparison of the performance between 20 MP and 50 MP camera configurations in which we assume imaging a $1 \times 1 \times 1$ mm volume. The reimaging rate for the whole dataset is roughly 10%, of which over 90% were caught by real-time QC embedded in the piTEAM pipeline and the remainder during post-processing or segmentation. The nondestructive nature of ssTEM and very cautious design of R2R GridStage allow us to reimage bad montages. We reimage the whole section, instead of targeting just specific tiles, as this facilitates the downstream stitching and alignment. This reimaging based on real-time QC was processed by batch, taking advantage of reliable random access to requested barcode, and was completed during the 6-month data collection period.

**Table 1 Performance metric during 1 mm³ production for 20 MP and 50 MP cameras.**

| Imaging parameters | Avg. production 20 Mpixel | Avg. production 50 Mpixel | 50 Mpixel peak | Unit |
|---|---|---|---|---|
| Frame size (pix) | 3840 | 5408 | 5504 | Aspect ratio of 1:1 |
| Resolution | 3.95–4 | 3.95–4 | 4 | nm/pixel |
| Tile overlap | 13 | 9 | 9 | % |
| Net imaging rate | 3.2–3.5 | 3.2–3.5 | 4.0 | Hz |
| FOV per side | 15 | 21 | 22 | µm |
| AOV | 225 | 468 | 484 | µm² |
| Pixels per image | 14.7 | 29.2 | 30.3 | Mpixel |
| Time per section | 30–40 | 15–25 | ~14 | Minute |
| Tiles w/overlap | 5100 | 2916 | 2600 | Per section |
| Sections per week (24/7) | ~300 | ~490 | ~700 | Section per week |
| Time to image volume with one scope @65% uptime | 900 | 525 | 450 | Day |
| Time to image volume with five scopes @65% uptime | 180 | 105 | 90 | Day |
| Completion of project in months @65% w/5 scopes | ~6 | ~3.5 | ~3 | Month |

An estimation of mean 65% uptime takes into account the downtime from microscope maintenance, barcode reading errors, and imaging pause determined by real-time QC.

**Imaging rate roadmap for 1 mm² montage per scope**

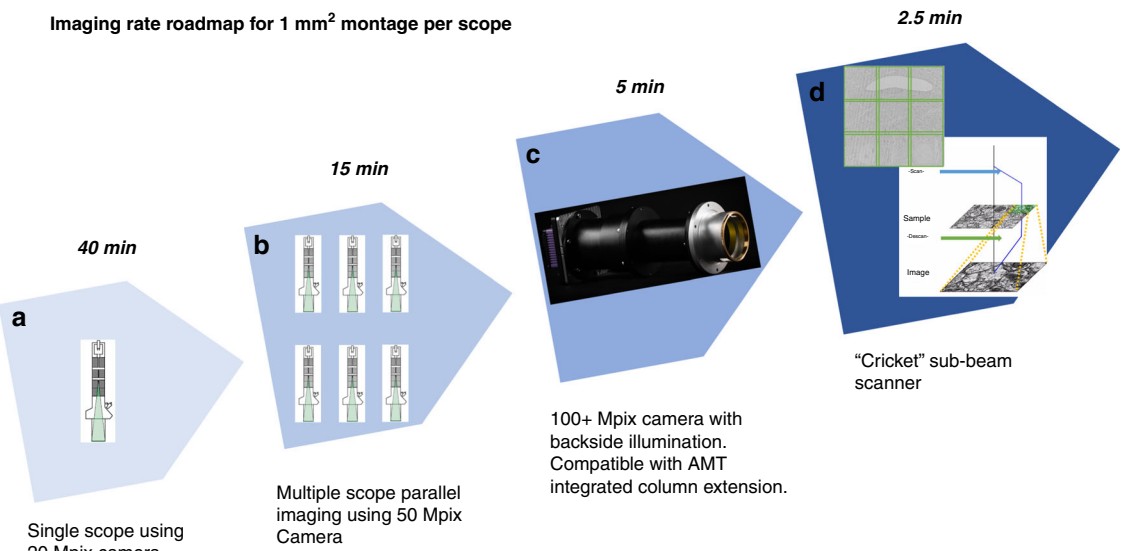

**Fig. 5 Imaging rate scaling roadmap. a** 1st generation platform using 20 Mpixel XIMEA camera; **b** 2nd generation platform with 50 Mpixel XIMEA camera (currently our platform contains six systems for parallel imaging); **c** AMT integrated column extension that can be combined with future commercial 100+ Mpix camera with back side illumination; **d** An example of Cricket which is a synchronous image sub-scanner and beam blanker. The inset is a 3 × 3 supertile acquired by Cricket sub-beam scanner. The electron beam is raster scanned to allow large fields of view to be quickly imaged.

## Discussion

The autoTEM/piTEAM imaging acquisition pipeline was designed to maximize imaging speed, robustness for imaging large datasets, scalability, and flexibility, while preserving easy upgradeability. Our current imaging pipeline uses 50 MPixel CMOS sensors (Fig. 5) that provide a large FOV per tile. The transition from 20 Mpixel cameras reduced the collection time for each 1 mm² montage from 40 min to 15 min (Table 1). We have simulated montage acquisition with the upcoming 100+ Mpixel CMOS sensors from major camera manufacturers (Fig. 5c) and estimated that the current imaging software and computing hardware could achieve a further 3× increase in speed (projected imaging metric in Supplementary Table 1). It is important to note that the performance described above was evaluated in a production setting where roughly one hundred million images were collected. In a research and development setting we have also tested several upgrades described below that provide either a

more compact design or further increases in imaging speed and scalability.

The early TEMCA systems[7] have an extended vacuum column to enlarge the final projected image. This increases the total height of the electron microscope from 2.4 m to 4.6 m, which is impractical for most facilities. We have experimented with an AMT ActiveVu lens assembly (CB500M-AV) that achieves the same magnification without the requirement of extending the height of the TEM. The lens we tested uses the XIMEA camera described above, so it was easy to integrate with our software. A High-resolution image taken by AMT lens and XIMEA 50Mpix camera is shown in Supplementary Fig. 10. In addition, the AMT integrated lens system is adaptable to future larger sensors to meet the requirement for scaling and serves as a good choice for the next generation column extension. Both new cameras and AMT integrated lens systems are drop-in replacements for our current autoTEMs.

Step-and-settle time of a sample stage consumes a large percentage of total image acquisition time per frame as the exposure time of the imaging system is reduced. The GridStage running in autoTEM consumes over 100 ms for a move to complete and trigger camera exposure. Such stage latency issues can be mitigated by using a hybrid mode that couples stage scanning with beam scanning and takes advantage of fast deflection of the electron beam, whose response time is measured in nanoseconds. We have tested a novel beam-deflecting device (Voxa's Cricket[TM]) that raster scans the electron beam to allow imaging multiple fields of view without moving a sample stage (Fig. 5d). This approach provides a wider field-of-view and decreased overhead. The beam sub-scanning system deflects the electron beam around the sample in a matrix pattern ($3 \times 3$), while scanning coils below the object plane de-scan the image onto the TEM camera. The sub-tiles obtained using this supertiling technique are acquired and stitched together to form a large composite image, in effect creating a virtual camera of larger size through electron-optics and computation. A 16 Mpixel image acquired at one position can, for example, be expanded to 120 Mpixel at that same stage position using a $3 \times 3$ supertile, thus drastically expanding FOV while eliminating 8 out of 9 stage movements.

We also paid attention to the affordability of any single machine, which was a primary concern about the original TEMCA systems[7,14,22]. Based on our design, if there is already an available TEM, a system that can achieve net imaging rates of at least 100 Mpixel per sec can be built with additional components that cost roughly $125,000. If a microscope is not available, refurbished systems can be modified as long as they come with ports for installing reel systems. For example, we purchased JEOL 1200EX-II for ~$125,000. This 1980s era microscope provides robust hardware and is abundant in the resale market. Such engineering approach makes a single autoTEM more affordable than other options for high-throughput EM and requires much less expensive facilities.

Finally, we are developing a suite of software modules to perform real-time lens correction and 2D montaging using the template matching results collected through image QC. This allows us to perform the 2D stitching of complete sections during acquisition using the graphics card of the acquisition computer and significantly save computation storage and time to do the image processing afterwards. We can also monitor the change in lens distortion over time and receive immediate quantitative feedback on the quality of images and stitched visualization. We are in the process of collecting a new volume dataset and comparing the performance with the existing post-imaging feature matching pipeline.

EM connectomics has seen remarkable advances in the last 10 years, making it poised to examine synaptic connectivity of neuronal networks at a very large scale. For this to happen, we took an industrialized approach to build an image acquisition pipeline, piTEAM. The entire EM imaging pipeline from sample transfer, image acquisition, and image QC is a continuous automated process. Constant feedback from all stages of the pipeline ensures data integrity and quality is maximized with minimal need for manual intervention. The transition from a prototype to an industrialized production pipeline was the most challenging problem. To facilitate this effort, we put a strong emphasis on standardization and consistency at every step of the pipeline, resulting in a suite of methods that are open and affordable.

The piTEAM approach allows for further increases in speed, both per microscope—by using increasingly large and fast cameras or with beam deflection, and per facility—by increasing the number of microscopes. In 2018, we used our six-microscope piTEAM platform to collect 2 PB of EM images of 1 mm³ mouse visual cortex at synaptic resolution over the course of 6 months. We anticipate the net average rate of a single microscope to increase from the 100 Mpixel per sec to 500 Mpixel per sec, through a combination of larger sensors and the beam-scanning. At this rate, a single microscope should be able to image a cubic millimeter in roughly 100 days. This throughput at the cubic millimeter range makes piTEAM ideal to investigate microcircuits across species, cortical regions in health and disease, in a framework that focuses on production of brain maps at large scale. Although this pipeline was designed for connectomics, any other field requiring automated serial-section imaging at the ultra-structural level can take advantage of the automation, high throughput, and affordability of the methods described in this manuscript. Our approach to TEM imaging can be used either as a single autoTEM or as a full piTEAM pipeline for distributed imaging that can be implemented both within a large dedicated facility like our own as well as distributed over a community of individual laboratories.

## Methods

**Tissue preparation and initial experimental setup**. All procedures were carried out in accordance with the Institutional Animal Care and Use Committee at the Allen Institute for Brain Science. All mice were housed in individually ventilated cages, 20–26 C, 30–70% Relative Humidity, with a 12 h light/dark cycle. Mice (CamK2a-tTA/CamK2-Cre/Ai93, CamKII-tTA/tetO-GCaMP6s, Slc-Cre/GCaMP6s) were transcardially perfused with a fixative mixture of 2.5% paraformaldehyde and 1.25% glutaraldehyde. After dissection, slices were cut with a vibratome and post-fixed for 12–48 h. Slices were extensively washed and prepared for reduced osmium treatment (rOTO) based on the protocol of Hua et al.[37] Ferricyanide was used to reduce Osmium and Thiocarbohydrazide (TCH) for further intensification of the staining. Uranyl acetate and lead aspartate were used to enhance contrast. After resin embedding, ultrathin sections (40 nm) were either manually cut in a Leica ultra-microtome or automatically onto GridTape using an RMC Automated Tape Collecting Ultramicrotome. The sectioning of 1 mm³ volume usually takes about 1 week. After sectioning, the samples are loaded into the autoTEM (~2 h.), and the microscope pumped down to achieve the vacuum level of 1E-7 Torr. The pump-down time depends on the number of sections being loaded onto the microscopes. For the large reels containing 5500 sections, this process can take up to 24 h. After vacuum is reached, we follow normal TEM operation routine to bring up the HT voltage and filament current and then align the beam. Calibration of the autoTEM involves tape and tension calibration for barcode reading, measuring beam rotation and camera pixels, and stage alignment. After which, imaging can start. These calibration procedures cost roughly a day, but are only required when changing the tape or filament.

**VOXA gridstage sprite**. The Voxa GridStage Sprite is used with sections collected on to metal grids. A 3D rendering is shown in Supplementary Fig. 1a. The stage combines two linear piezo stages to have $x$ and $y$ translation. The axes of the stage have a scan resolution of ~1 nm with repeatability of ~50 nm. The system was designed to translate Grid Sticks, which are cartridges that can hold up to 16 standard 3 mm TEM grids. Since Grid Sticks are much larger and more robust than individual TEM grids, sample handling errors were reduced while increasing the load density. An additional advantage is that Grid Sticks are a storage medium for standard sample grids, allowing for easy indexing when archiving thousands of sample grids.

**autoTEM column extension**. The autoTEM column extension continues the vacuum column of the microscope and is terminated with a custom scintillator coated with P43 phosphor, followed by a custom leaded glass window that blocks X-rays from escaping through the bottom of the column. The rest of the column extension is encased in lead shielding panels to block X-rays around the column. Below the column is a custom camera housing with a single camera (XIMEA CMOSIS CMV20000 or CMV50000) that images the scintillator screen. The PCI-E interface offered by the CMOSIS camera also provided sufficiently fast data transfer rates to enable on-the-fly GPU-based image processing and quality control.

**TAO database and automatic ROI generation**. Given that tissue sections can number in the tens of thousands for large-scale datasets, mapping tissue ROI becomes a labor-intensive task and requires an automation process as well. Therefore, TAOs (TEM Acquisition Objects) define the ROI to be imaged for each aperture and are automatically created from the optical images captured from the ATUM system during sectioning. TAOs are uploaded to a cloud database upon creation. This allows users to create a highly standardized datasets and to swap sections across microscopes without the need for manual calibration, ROI setup, and validation.

We have also developed automated scripts to create tissue ROIs. The tissue area on each aperture can be detected automatically from optical sectioning images (Supplementary Fig. 6). Based on pre-defined rules of absolute distance to the tissue corner along with tissue compression scaling factors, we achieved ~97% success rate of placing correct ROI's. Once data are uploaded to cloud database, it is immediately available for use in TEM imaging or QC analysis. We have verified through 2D montage that the ROI placement is sufficiently precise using this method.

The present automatic ROI generation sequence per aperture is as follows: (1) extract the barcode, (2) threshold the image to find the aperture candidates, (3) select the largest aperture candidate, which is nearest the centerline of the optical image, (4) from the aperture extract the centroid and bounding box, (5) find the tissue area using HSV color segmentation over a range of colors, (6) filter out noise in the tissue area by removing small interior blobs and trimming small outer tendrils, (7) reject tissue candidates which don't meet a minimum size criteria and extract the centroid of the tissue area, (8) optionally create a fixed size ROI at the tissue centroid, (9) insert all extracted data in to the TAO, and upload the original image using the specimen_id, media_id, and barcode as keys for retrieval. If suitable tissue cannot be automatically located, an empty TAO placeholder is created and uploaded for eventual manual ROI definition.

In addition to the automatic ROI generation described above, pyTEM GUI is also a multi-resolution, web-based CAD system for manual ROI definition and editing at different EM magnifications or on optical images (Supplementary Fig. 3). Software can also automatically locate the ROI corners for users to refine ROI placement, which may be required to correct for beam hysteresis when switching magnifications. During montage runs, a user specifies a range of ROIs to image using a format similar to Microsoft Word page print selection (1–8, 13, 783–799).

**Camera calibration**. The nature of an electron beam is dependent upon the characteristics of the electron-generating filament and its alignment within the system. Thus, after initial installation and whenever a TEM filament is changed, the imaging subcomponents of the system need to be recalibrated. The most critical features to tune are the pixel resolution (nm/pixel) and the angular beam rotation. As part of the real-time QC module, the pixel calibration is done through template matching. The image pixel displacements of a $3 \times 3$ grid of template located at the center of the screen are measured for known stage displacements. The standard deviation of each $X$ or $Y$ vector is less than one pixel and results are now highly repeatable. Overall, this method ensures calibration consistency across multiple autoTEM systems.

**Aperture centroid finding**. The coordinate system used to place ROIs on the optical ATUM image assigns the aperture centroid as the (0, 0) reference point. ROIs are then defined in physical distance units as X/Y offsets from this centroid point along with a rotation angle. When an aperture is imaged in the montage acquisition magnification (4 nm/pixel), the tape subsystem first positions the aperture in the approximate center of the column. Next, the centroid of the aperture is detected in stage coordinates using a binary search algorithm for the light–dark transition point demarking each of the four aperture edges: top, bottom, left, right. Each ROI is then translated to this stage coordinate centroid, and further rotated by a magnification-specific beam rotation which is measured during the calibration process.

**Brightfield and darkfield correction**. To compensate for camera, lens, and illumination non-linearities, each montage image is corrected using previously acquired brightfield and darkfield images. The darkfield corrects sensor per-pixel dark current offsets and is acquired once per week, or whenever the camera gain is altered. The brightfield (example in Supplementary Fig. 7) corrects illumination gradients (which can change with electron beam drift), camera lens non-linearities, and sensor per-pixel gain variations. A new brightfield is acquired for each aperture.

The darkfield is acquired by averaging together 16 images with the EM beam turned off. The brightfield is acquired over tissue with the electron beam on while randomly moving the stage slowly across a 300 μm radius region, integrating light from light and dark tissue regions. Sixteen such integrated images are averaged and then scaled to create the brightfield.

For each montage image, a corrected image is created using the brightfield and darkfield images on the GPU in floating point format:

$$\text{corrected} = (\text{image} - \text{darkfield})/(\text{brightfield} - \text{darkfield}). \quad (1)$$

The corrected image is converted to 8 bpp and saved as a TIFF file.

**Auto focus**. Over the course of scaling montage area from hundreds of μm$^2$ to several mm$^2$, the autofocus algorithm had to evolve to deal with a property of the JEOL 1200EXII TEM where overall image intensity changes with focus value. The current focus measurement is derived as follows (Supplementary Fig. 8): (1) from the corrected $3840 \times 3840$ or $5408 \times 5408$ image, extract the center $2048 \times 2048$ pixels, (2) compute the log magnitude of the Discrete Fourier Transform (DFT):

$$\log\left(1 + \sqrt{\text{Re}(\text{DFT}(I))^2 + \text{Im}(\text{DFT}(I))^2}\right) \quad (2)$$

(3) create an image mask, which excludes lowest 6 frequency components and high frequency components above 1600, (4) perform a polar to rectangular transformation of the DFT components within the mask area, (5) find the average of the components for each row, (6) sum the averages. Discarding the highest frequency components (which are largely noise) makes the algorithm less susceptible to the image intensity changes that happen with the change of focus. Determination of optimal FFT frequency components was derived experimentally by comparing FFT scores at different exposures and different frequency components.

Through the experiments from pilot datasets, we noticed that a single EM focus point at the ROI centroid was not sufficient for a large 1 mm$^2$ montage. We sometimes failed to derive optimal focal point when a blood vessel or other void existed at the ROI centroid; or when there was a sample height gradient across ROI due to tissue section being placed toward the aperture edge. To improve the autofocus algorithm, we now perform focus optimization at additional satellite points distributed across the ROI, averaging the remaining measurement points to create a reasonably optimal focus value for the whole montage.

**Performance tuning (LaB6 filament, stage performance)**. To achieve the optimal speed and quality of the image platform, we have also made improvements in terms of electron source, camera lens, and stage performance. The electron source for the autoTEM system is a critical component for long-term imaging experiments. Traditionally the JEOL 1200EXII leverages both Tungsten hairpin style filaments and Lanthanum hexaboride crystal (LaB$_6$) as possible electron sources. Tungsten filaments, although easier to operate, demonstrated poor stability for long periods of time due to deterioration of the tungsten source, as well as a short life span (~100 h). LaB6 filaments have a more stringent ultimate vacuum pressure but have a much larger current density (higher electron flux per unit area), which reduces exposure times for the capture camera as well as a much longer lifetime (1000–2000 h). The average lifetime by using LaB6 filaments during production imaging was about 1 month for continuous 24/7 operation. In addition, the exposure time using the tungsten source was 100–150 ms as compared to 50–80 ms using LaB6, with same camera, optical arrangement, and specimen.

Among the 300 ms acquisition cycle for each tile, the stage step-and-settle time from initializing the stage move to finishing the move and send the complete status back to pyTEM is averaged at ~120 ms, consuming almost half of the total cycle time. Therefore, it is very important to optimize the stage performance parameters. We focused on stage speed, acceleration, and dwell time to minimize stage settling time. In general, we have found a linear correlation between stage acceleration and speed to the cycle time: the larger the acceleration, the less the step-and-settle time; the higher the speed, the less the step-and-settle time but it caps around 10 mm s^−1. An appropriate dwell time is also necessary, otherwise the random streaks of blurry tiles appear on the montage, because of insufficient wait time for stage to settle for camera exposure before the next move. Larger frame size also requires more dwell time because of increasing step distance for each tile.

**Software**. The *pyTEM* system is comprised of Python modules, which orchestrate automated acquisition. The software is built upon a state machine where each state represents an acquisition step. The *TEM Graph*, which is at the core of the acquisition and real-time processing pipeline is written in in C$^{++}$ using OpenCV and mostly running on acquisition computer GPU. Analytics of the different sensors composing the Multi System Monitoring were visualized with Grafana.

**Image processing after acquisition**. For all post-processing stitching, we used the HHMI renderer framework (https://github.com/saalfeldlab/render10). This approach does not write intermediate renderings to disk, instead calculates and stores the transforms while maintaining the original raw data. We apply the following transformations to stitch tiles: (1) for lens distortions the tile images are corrected by computing a nonlinear transformation[38]. The technique involves computing point correspondences for each pair of images that overlap with each other within a section. These point correspondences are utilized to find the geometric transformations that would correct for the lens distortion effects; (2) The above technique is also used to stitch the images that overlap with each other within a section. The real-time QC from the imaging pipeline ensures a minimum of 7% overlap from the neighboring tiles for downstream 2D stitching. The advantage of such an approach is its capability to compute a variety of transformations that can produce a high-quality stitching. These transformations range from a similarity transformation to a higher order polynomial transformation.

**Reporting summary**. Further information on research design is available in the Nature Research Reporting Summary linked to this article.

## Data availability

Electron microscopy data that supports the findings of this work are available on http://www.microns-explorer.org and on https://github.com/AllenInstitute/piTEAM

## Code availability

We will share software, bill of materials, and hardware design drawings upon request. Software will be shared under the Allen Institute Software License and Contribution Agreement, subject to any applicable third-party licensing restrictions. Bill of materials and hardware design will be shared under the Allen Institute Terms of Use: https://alleninstitute.org/legal/terms-use/ and deposited on https://github.com/AllenInstitute/piTEAM.

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

## Acknowledgements

We thank our project manager Shelby Suckow for her exceptional work on managing the collaboration and for keeping us aligned and on time. We thank David G. C. Hildebrand, Aaron Kuan, Jasper Maniates-Selvin and Logan Thomas for developing, milling, advice, and assistance on the use of GridTape; Cliff Slaughterbeck for engineering project Management and engineering design review. T. Ayers, R. Smith and Luxel Corporation for coating tape; JoAnn Buchanan for histology; Joe Mancuso and Adam Manganiello for providing the AMT extended column and assisting experimental data collection; Davi D. Bock for his advice on TEMCA column extension and his thoughtful comments on the paper; Agnes Bodor for feedback on image quality; Lawrence Own and Teddy Derego for their development and continuous support in GridStage and GridCon software; and Sebastian Seung and Adrian Wanner for discussion on imaging strategies and improvements. We thank Yang Li for creating the videos with the EM montages. The datasets described above were previously imaged with 2P-calcium imaging by the lab of Andreas Tolias and subsequently fine aligned and segmented by the lab of Sebastian Seung. This work was supported by the Intelligence Advanced Research Projects Activity (IARPA) of the Department of Interior/ Interior Business Center (DoI/IBC) through contract number D16PC00004; NIH Grant R21NS085320 (W-C.A.L.); and by the Allen Institute of Brain Science. The U.S. Government is authorized to reproduce and distribute reprints for Governmental purposes notwithstanding any copyright annotation thereon. The views and conclusions contained herein are those of the authors and should not be interpreted as representing the official policies or endorsements, either expressed or implied, of the funding sources including IARPA, DoI/IBC, or the U.S. Government. We thank the Allen Institute's founder, Paul G. Allen for his vision, encouragement and support.

## Author contributions

N.M.C. and R.C.R. conceptualized the project. W.Y. conducted process integration, experiments and system validation for the imaging pipeline. D.B. and M.S. designed and built the autoTEM and closed-loop system. J.B. and D.B. designed, built and developed pyTEM software for image acquisition. J.B., W.Y. and D.B. developed real-time QC. W.Y., D.B., D.J.B., M.T. and N.M.C. performed imaging and troubleshooting of 1 mm³ dataset for mouse visual cortex. R.M.T. was responsible for data transfer and storage. D.B., D.W., J.P. and A.B. developed the prototype pyTEM image acquisition software. C.O. and M.M. built and developed the reel-to-reel GridStage and GridCon software. D.K. and J.B. developed real-time lens correction and 2D montage. G.M., R.M.T. and D.K. performed stitching and alignment. D.C., D.R. and C.F. provided manufacturing process engineering support. W-C.A.L. and B.J.G. developed and provided a prototype reel-to-reel TEM imaging stage and tape handling machine that was used during the development stages of the pipeline, they also provided materials, advice and assistance for the use of GridTape. W.Y., D.B., J.B., C.R. and N.M.C. wrote the paper with input from all authors.

## Competing interests

Harvard University has filed patent applications regarding GridTape (WO2017184621A1) and the prototype reel-to-reel TEM imaging stage (WO2018089578A1) on behalf of the investigators (B.J.G., W-C.A.L.) and others. C.S.O. and M.F.M. have a financial interest in Voxa. The remaining authors declare no competing interests.
