## [Peer Review File · Nature Communications]

Reviewers' comments:

Reviewer #1 (Remarks to the Author):

In their manuscript entitled "A Petascale Automated Imaging Pipeline for Mapping Neuronal Circuits with High-throughput Transmission Electron Microscopy" the authors describe an approach to acquire massive amounts of serial sections by TEM for connectomics research. They have decided to split the publication of their technological advances into two manuscripts with the one presented here focusing on the acquisition, quality control and data handling strategies.

Recommendation:

The technological advances described in the manuscript are impressive in opening up new scales of TEM imaging. I can therefore support publication, given improvements regarding the following aspects.

Major comments:

In general, there is a fair amount of repeating information in the manuscript. In my view, the article could be boiled down significantly yielding a concise description of the achievements. A better separation of the core technological improvements with the resulting benefits in throughput on one side (results part) and the specific technological details (methods part) on the other would help improving the comprehensiveness.

Stronger focus on the acquisition work flow, QC and software: Since the technology to supply the specimen to the microscope (via tape) is described in a separate publication, its description should be reduced to the minimum necessary. This will also help avoiding to read like advertisement for the respective commercial products.

Discussion of relevant (dis-)advantages, more insights into the operational routines:

cost-effectiveness: Please comment on the manpower necessary to

- 1) install and get a similar setup operational
- 2) operate an array of microscopes (engineer, software engineer)
- 3) prepare the specimen (in the tape case)

Effect of QC on imaging: How often is re-imaging typically triggered? Are there cases of specimen issues that cause repeated issues (charging,...)? Will acquisition of such problematic tiles be dropped eventually?

It does not become clear if, how and at what steps image data is altered/processed by software. What transformations are applied to stitch tiles, how is the volume combined from the tiles, what distortions appear? The last paragraph of the "discussion" part introduces future solutions; please also include how these challenges are dealt with in the presented setup.

Statements on overall success rate:

This manuscript mainly focuses on the increase in data acquisition. I am missing statements on the overall success rate of the entire presented work flow, from the en-bloc specimen to the reconstructed 3D volume. What is the loss-rate of sections? How many tile acquisitions fail due to damage/contamination/charging?

Comparison with Volume SEM techniques:

The manuscript almost entirely lacks a comparison with established Volume EM methods. Since these are very commonly used in connectomics, it is key for the community to be presented with a thorough comparison with these approaches not only in terms of data rate and instrument costs.

Have the authors compared a preparation of neuronal circuits of the same source that has been imaged using volume SEM techniques? What is the effect of compression when using the tape pickup and such thin sections?

Better structuring the manuscript:

The last big paragraph of the introduction already comprises many of the key results.

Figures:

- The panels in Fig.1, 2 and 4 appear in very confusing order
- The references to the figure panels in the main text should be more explanatory.
- Figures 2 and 3 could be combined into one panel illustrating the procedure and another parallel panel describing the individual modules. Fig.3 contains way too many technical details.
- Fig. 4. Here, the caption contains a lot of information that should rather appear in the main text. The information in the main text is hardly comprehensible (p.15 "If lens calibration..."). Here, separating the operational principles from technical details might help a lot.
- axes in Fig. 5 d/e are not readable. The conclusions of these measures should also be more clear (main text)
- Fig. 6d, the caption contains information that should appear in the main text/discussion.

Minor comments and questions:

- are all the TEM systems incorporated in the array of the same model? What are the prerequisites to run the control software and install the tape-reel stage?
- introduction, last paragraph: 24/7 imaging with an average uptime of 65% is mentioned. Please elucidate on the problems and possible solutions. What is the time loss due to the necessary monthly swap in Lab6 sources and re-alignments/calibrations?
- p. 10 last paragraph: "To achieve ..., TEM Graph is written ...": This sentence makes no sense.
- p.12 "2) seeking barcode": How is this achieved in the TEM stage? Is there a special sensor/reader incorporated?
- p. 16 stitching: Please include more technical details in the methods part to illustrate what type of transformations were used, how lens and specimen distortions were handled, etc.
- p. 19 height of the microscope: please use SI units

Reviewer #2 (Remarks to the Author):

To the authors -

The authors describe an automated image acquisition pipeline using multiple TEMs for imaging serial sections of a mouse brain volume of 1 mm³/ 40 nm sections.

General comments:

The EM-field is lagging behind other methods for high throughput analysis. Several approaches are being explored by some labs, but in >99% of publications only single snap shots of typically selected areas are shown. Only ~1% of labs work to get high-content EM done, and the current manuscript is an excellent example, building on previous work of Brock et al., using custom re-engineered microscopes (that actually were built to take analog images, not digital) to allow automated EM of large areas of interest. The automated workflow is well developed with respect to hardware and software and may help to set the stage for future routine EM by many.

The downside of the approach is that the application likely will be very limited because of the need of multiple TEMs of the same type which have to be re-engineered, which makes it a not very accessible approach for most EM facilities. Moreover, other ways to image 1mm³ brain tissue have already been described, but only the disadvantages of these approaches over piTEAM are discussed, not their huge benefits. The discussion should be more balanced.

Specific comments & questions:

- Abstract/ First sentence: 'Serial-section electron microscopy is the method of choice for studying cellular structure and network connectivity in the brain.' This exemplifies how biased the authors are about the technique, and dismisses 3-view block face method. They might have the opinion, but serial-section EM is a method... . Balance on other approaches is needed throughout the manuscript >6 occasions.
- Restructure the text to prevent duplications.
- Readability can be improved by fewer abbreviations and reducing numbers (e.g. make use of another table).
- Automated acquisition after initial experimental setup (p4): how much time is needed for 'initial experimental setup'?
- 'TEMs achieve perhaps the highest signal-to-noise ratio¹⁰, especially for fast imaging, but commercially available TEMs are neither designed nor optimized to efficiently image serial sections at a large scale.' Compared to what?
- Page 5 is redundant and can be completely deleted.
- P6. Inches? Later feet? Please use metric system.
- Two stages mentioned: Voxa GridStage Sprite is not used to obtain data shown in this paper. Only the tape version is used. Why mention this?
- Many abbreviations make the text not easy to read (MSM TAO AWS CLAHE....) PiTEAM and pyTEM is confusing. Please use less uncommon abbreviations.
- Another example of redundant text that can be deleted for clarity of the manuscript: P8. 'In order to collect a petascale data set over months of continuous imaging, we designed the autoTEM system to be almost completely autonomous. In the sections below we describe the infrastructure that enabled this automation starting with the software.'
- And another one: 'In summary, the image acquisition software has been designed with the following goals in mind:
 - Completely autonomous operation. An autoTEM system should be able to run without user intervention beyond initializing the experiment.
 - Adaptable to advances in sensor technology, enabling fast and affordable scaling. We have developed an abstracted software architecture, functional validation test plan, and change management plan to understand the impacts of sensor upgrades on image quality and effective throughput.
 - Automatic ROI generation from optical images or low-mag EM images.
 - Remote control of the TEM machines using web technologies.
 - A cloud-based database, associating specimens, tapes, apertures, and ROIs which span optical and EM imaging modalities. This database manages the millions of images that are generated and tracks them back to ROIs, section ID, microscope configuration, imaging condition, etc.
 - Remote multi-system monitoring (MSM) tool that analyzes system and environmental variables (such as temperature and vacuum, among others) and uses feedback mechanisms
- I did not go into all detail in all the suppl. Data. Many of which seems redundant, exemplified by Fig.S4.
- P14 Figure 4b should show deviation of expected tile position by colors. Very poorly visible. Please enlarge figure 4b.
- P15. Fig 4e shows a fairly uniform focus map with tissue structures such as blood vessels being highlighted. This suggests that less homogeneous tissues (like pancreas, intestine) autofocus might fail. Also additional satellite points (p 41) will not solve this. This will limit this approach to 'homogeneous' tissues like brain tissue, whereas the authors claim it is also suitable for other tissues (on p21).

- Also the need of matching features (p15) for stitching limits this approach to a limited set of tissues. Tissues with (relatively) large areas which are poor in features (like lung, thyroid gland etc) will fail in this workflow.
- P18. Upgradeability: Increasing amount of Pixels of CMOS sensors will at some point not necessary result in higher resolution images, because the details in the image on the phosphorescent screen will be limited. Especially for other approaches where ultrastructural details are important in contrast to brain connectomics where only heavily stained membranes are of interest. With the current setup in dataset very little details can be distinguished other than membranes as can be seen in the blow up in fig 5
- P19. Authors state their approach is superior to Multibeam SEM blockface imaging because their burst rate is a bit higher. However they do not mention that in a SEM with a slice and view approach the z resolution is twice as high. Also the stability of a FEG in a multibeam system is more stable compared to a LaB6 in a TEM and also has an approx. 20 times longer lifespan (~20 months vs 1 month) which is of high relevance if acquisition times are in the range of multiple months.
- P19. Please use metric units (15 feet)
- An issue not mentioned by the authors is image analysis. How to analyze petabytes of images? An additional algorithm to extract the relevant data and discard non relevant data should be very relevant. So on the fly building the connectomes for example.
- If QC fails over and over because something went wrong with the section, how will this be dealt with?
- New results on the XIMEA are resented in the discussion. Please restructure, put these into the results.

Below we address specifically all the reviewers' comments.

Reviewer #1:

1. In general, there is a fair amount of repeating information in the manuscript. In my view, the article could be boiled down significantly yielding a concise description of the achievements. A better separation of the core technological improvements with the resulting benefits in throughput on one side (results part) and the specific technological details (methods part) on the other would help improving the comprehensiveness. Stronger focus on the acquisition workflow, QC and software: Since the technology to supply the specimen to the microscope (via tape) is described in a separate publication, its description should be reduced to the minimum necessary. This will also help avoiding reading like advertisement for the respective commercial products.

Reply: We have made changes to the *Results* part to highlight our innovations in hardware development and software novelties (including the improvements to the reel-to-reel stage). The unique *hardware modifications* in piTEAM include:

- Single large CMOS camera and enlarged scintillator mimicking 12'' industrial Si wafer.
- Bidirectional section translation.
- Random access apertures through barcode reading. (Integrated barcode reader)
- Clamping mechanism, which caused a major failure mode: tape slippage during development.

We have also reconstructed the software section, to be more specific about the pipeline architecture as well as key components. More detailed functional features are provided in the *Methods* as suggested by the reviewer. The closed-loop imaging workflow including montage setup and run, error handling and real-time QC are addressed in the third section.

2. Discussion of relevant (dis-)advantages, more insights into the operational routines: cost-effectiveness: Please comment on the manpower necessary to:

2.1. install and get a similar setup operational

Reply: After a TEM system that matches the requirements for an autoTEM microscope is installed at a facility (see requirements) a team of three can accomplish the installation of the remaining components. Installation of microscope hardware can be accomplished by two individuals for one week. The software installation can be done by a skilled software technician. Finally, an integration test should be performed by the end user(s) to validate the systems' readiness, often with the aid of the software personnel.

2.2. operate an array of microscopes (engineer, software engineer)

Reply: The imaging pipeline has built-in automation and system control feedback to keep microscopes running by themselves 24/7. In general, it takes about 30% of an engineer's time during production to perform following tasks: error state recovery; QC, trouble-shooting scopes, tape and stage issues. The system calibration before each dataset usually takes a half day for each system. The filament change and alignment take a couple of hours, occurring every 1~2 months. The tape loading takes about a few hours. Its frequency depends on tape length. For the 1 mm³ dataset, a tape of 5500 sections can run through the whole data collection cycle of ~ 6 months. We later imaged another 1 mm³ dataset and it took an average of 2 months to complete a tape of 2500 sections. We loaded a second tape on to each system in the middle of production. More efforts were spent on hardware and software development in-between the dataset production and validation, but we don't expect other end users to repeat them.

2.3. prepare the specimen (in the tape case)

Reply: Tissue preparation (histology and embedding) is not specific for the tape collection or requires human resources compared with other sectioning approaches. The shape of the tissue block is specific to the tape method, but the manpower and time required is similar to any other method and dependent on block thickness (for 1 millimeter it takes ~1 week/1 person to be done perfectly and to minimize risk of damage). Finally, it takes between 17-26 seconds per section when cutting.

3. Effect of QC on imaging: How often is re-imaging typically triggered? Are there cases of specimen issues that cause repeated issues (charging...)? Will acquisition of such problematic tiles be dropped eventually?

Reply: The reimaging of the 1 mm³ dataset described in this manuscript accounted for about 10% of the total number of sections. This was the first time the imaging pipeline was used for production, and we saw reimaging rate dropped to about 5% for the second 1 mm³ dataset collected in 2019. We expect this number to go down further in the future with more improvements along the way. The built-in QC of imaging pipeline captured over 90% of the total sections required for reimaging and drastically save the back and forth and delay for image processing. We usually reimage any section that failed QC every 200 sections, mainly to minimize excessive tape

movement which increases the risk of section damage. Reimaging is done by batch, another benefit from barcode random accessibility.

We rarely encounter charging issue during TEM imaging due to good grounding through tape and carefully chosen materials in the GridStage. Charging was once a major failure mode during development and caused tile blurriness but was solved later by modified circuit and tape encapsulation. The error was caught by focus map in QC.

Currently the top recurrent failure modes are intermittent beam intensity variation (at the end of filament life), intermittent focus deviation due to stage error or vibration, barcode reading error. All sections experienced those failures were reimaged.

We don't drop specific bad tiles, but instead reimage the whole section to make the downstream stitching and alignment much easier, and it is also faster to reimage the whole section instead of identifying locations of bad tiles. Because of the non-destructive nature of ssTEM and the very cautious design of R2R GridStage, we were able to reimage a vast majority of those bad montages, and only saw rare cases of damaged sections for reimaging.

4. It does not become clear if, how and at what steps image data is altered/processed by software. What transformations are applied to stitch tiles, how is the volume combined from the tiles, what distortions appear? The last paragraph of the “discussion” part introduces future solutions; please also include how these challenges are dealt with in the presented setup.

Reply: The raw data is not transformed by the image acquisition pipeline. For all post processing stitching we used the HHMI renderer framework (Zeng et al, 2018, <https://github.com/saalfeldlab/render>). This approach does not write intermediate renderings to disk, instead calculates and stores the transforms and we always keep the original raw data.

We apply the following transformations to stitch tiles:

- a. The tile images are corrected for lens distortion effects by computing a non-linear transformation (Kaynig et al. 2010). The technique involves computing point correspondences for each pair of images that overlap with each other within a section. These point correspondences are utilized to find the geometric transformations that would correct for the lens distortion effects.
- b. The above technique is also used to stitch the images that overlap with each other within a section. The advantage of such an approach is its capability to compute a variety of transformations that can produce a high-quality stitching. These transformations range from a similarity transformation to a higher order polynomial transformation.

The details are now included in the *Methods*.

Zheng, Z. et al. A Complete Electron Microscopy Volume of the Brain of Adult *Drosophila melanogaster*. Cell 174, 730-743.e22 (2018).

Kaynig, V. et al, Fully Automatic Stitching and Distortion Correction of Transmission Electron Microscope Images, Journal of Structural Biology, 171(2), 163-173 (2010).

5. Statements on overall success rate. This manuscript mainly focuses on the increase in data acquisition. I am missing statements on the overall success rate of the entire presented workflow, from the en-bloc specimen to the reconstructed 3D volume. What is the loss-rate of sections? How many tile acquisitions fail due to damage/contamination/charging?

Reply: No specimen was lost during en-bloc staining. The loss-rate of section is about 0.2% during ATUM sectioning and 0.1% during imaging due to non-manufacturing errors. The computational steps of stitching, alignment and segmentation did not cause section loss. The major failure modes of the acquisition and reimaging rate was described above in the reply to reviewers comment 3 “Effect of QC on imaging”.

6. Comparison with Volume SEM techniques. The manuscript almost entirely lacks a comparison with established Volume EM methods. Since these are very commonly used in connectomics, it is key for the community to be presented with a thorough comparison with these approaches not only in terms of data rate and instrument costs. Have the authors compared a preparation of neuronal circuits of the same source that has been imaged using volume SEM techniques?

Reply: The authors have added a more detailed review and comparison of the state-of-the-art EM methodologies in the *Introduction* on page 3-4.

We have compared with published EM images taken by other methods and believe that our dataset demonstrates comparable high image quality. Moreover, this high quality is sustained over millions of tiles. We have not collected the data from the same source imaged using volume SEM, but have a plan to experiment some sections using Multibeam SEM facilities in Janelia HHMI. However, due to the current circumstances and special timing, we will not be able to complete this additional experiment in a timely manner.

7. What is the effect of compression when using the tape pickup and such thin sections?

Reply: The effects of compression when using the tape pickup are similar to the effects described by other serial section electron microscopy methods (15% to 30%, reviewed by Kubota *et al*, 2018) as the compression depends mostly on sample preparation, resin and type of knife and less on the method of pickup. To address this deformation, before sectioning we image the sample with X ray microCT, and then co-register this dataset with the EM data to allow us to correct for compression.

Kubota, Y. *et al* Large Volume Electron Microscopy and Neural Microcircuit Analysis. *Frontiers in Neural Circuits*. <https://doi.org/10.3389/fncir.2018.00098>. (2018)

8. Better structuring the manuscript:

8.1. The last big paragraph of the introduction already comprises many of the key results.

Reply: We have removed most of the last page of *Introduction* and substituted it with a light version of the Summary.

8.2. Figures:

8.2.1. The panels in Fig.1, 2 and 4 appear in very confusing order. The references to the figure panels in the main text should be more explanatory.

Reply: We have rearranged the orders in Fig.1, 2, 4 (# 3 in the revised manuscript). The figure captions have also been updated.

8.2.2. Figures 2 and 3 could be combined into one panel illustrating the procedure and another parallel panel describing the individual modules. Fig.3 contains way too many technical details.

Reply: We have combined Fig. 3 into Fig. 2 as panel d. Fig. 3 provides a comprehensive illustration of the key element in the pipeline: closed-loop imaging workflow. The authors would like to keep the graph diagram, which will help audience to understand the workflow better.

8.2.3. Fig. 4. Here, the caption contains a lot of information that should rather appear in the main text. The information in the main text is hardly comprehensible (p.15 “If lens calibration...”). Here, separating the operational principles from technical details might help a lot.

Reply: Figure 4 is renumbered as Figure 3. We have rewritten part of the real-time QC to highlight the principles and key results and removed confusing sentences. The details about the template choice and overlap are covered in *Methods* and figure caption.

8.2.4. axes in Fig. 5 d/e are not readable. The conclusions of these measures should also be more clear (main text)

Reply: The fonts and axis of Fig. 5d (4d in the revised manuscript) have been fixed and the inset of panel e is removed since the bar chart provides the similar information as histogram. We also included more explanations of those measurement results in the main text.

8.2.5. Fig. 6d, the caption contains information that should appear in the main text/discussion.

Reply: The information has now been moved to the main text/discussion.

Minor comments and questions:

9. are all the TEM systems incorporated in the array of the same model? What are the prerequisites to run the control software and install the tape-reel stage?

Reply: For consistency of imaging results we used the same TEM model (JEOL 1200EXII) in all our systems. There are a few requirements for a TEM to be compatible with the equipment. Installing the reel-to-reel system requires a large gap objective pole (recommended width: 18 mm; height 16 mm; gap: 11 mm) with ports opposite of each other across the pole piece. The installation of the imaging system requires a bottom camera or film chamber port to attach the extended column, this would require machining to connect the column based on the microscope model. The software requires an interface with the microscope (the JEOL 200EXII has a serial card) to control the magnification, beam size and shift. This feature will be vendor specific.

The software runs on a custom acquisition computer running Windows 10. We determined the computer hardware required consists of a Nvidia graphics card to take advantage of CUDA GPU compute operations that is leveraged by OpenCV graph. In addition, a fast network connection (10Gb/s NIC) and a fast storage RAID (SSDs in RAID0) facilitate data storage and transfer to keep up with acquisition.

10. introduction, last paragraph: 24/7 imaging with an average uptime of 65% is mentioned. Please elucidate on the problems and possible solutions. What is the time loss due to the necessary monthly swap in Lab6 sources and re-alignments/calibrations?

Reply: The major sources of downtime (35%) are roughly divided into the followings:

1. Barcode reading failures during sample translation: 15%
2. Microscope downtime such as HT failure, vacuum failure: 15%
3. Filament change including alignment: 5%
4. Camera calibration is usually done before imaging of a new tape starts (every 5500 apertures) and is also checked on a monthly basis: 2 hours
5. Tape calibration time cost is minimal, each calibration takes a few minutes and is also done before imaging of a new tape starts.

11. p.10 last paragraph: “To achieve ..., TEM Graph is written ...”: This sentence makes no sense.

Reply: The authors have restructured Software infrastructure to make it more concise. TEM Graph is the core of real-time image process, and now covered in Box 1 on page 8-9.

12. p.12 “2) seeking barcode”: How is this achieved in the TEM stage? Is there a special sensor/reader incorporated?

Reply: There is a barcode reader sensor on the stage that reads the barcode associated with each section. This is now better described on page 7.

13. p. 16 stitching: Please include more technical details in the methods part to illustrate what type of transformations were used, how lens and specimen distortions were handled, etc.

Reply: We have now added information to the *Methods* section (see also reply to comment #4)

14. p. 19 height of the microscope: please use SI units

Reply: We now use SI units.

Reviewer #2

General comments:

1. The EM-field is lagging behind other methods for high throughput analysis. Several approaches are being explored by some labs, but in >99% of publications only single snap shots of typically selected areas are shown. Only ~1% of labs work to get high-content EM done, and the current manuscript is an excellent example, building on previous work of Brock et al., using custom re-engineered microscopes (that actually were built to take analog images, not digital) to allow automated EM of large areas of interest. The automated workflow is well developed with respect to hardware and software and may help to set the stage for future routine EM by many.

The downside of the approach is that the application likely will be very limited because of the need of multiple TEMs of the same type which have to be re-engineered, which makes it a not very accessible approach for most EM facilities. Moreover, other ways to image 1mm³ brain tissue have already been described, but only the disadvantages of these approaches over piTEAM are discussed, not their huge benefits. The discussion should be more balanced.

Reply: The setup of multiple TEMs is not confined to a single model. The one we deployed is the JEOL 1200EXII. However, similar adaptation can be done on other JEOL models or FEI TEMs as long as the scopes provide the ports to host reel-to-reel housings. Any approach aims to scale electron microscopy horizontally by increasing the number of microscopes will likely use microscopes of the same type for the convenience of obtaining similar images, we don't see this as a downside of the microscopes that use. We also would like to point out that to replicate our pipeline is relatively straightforward, and the microscopes don't have to be extensively re-engineered, as the most engineering effort was done during our development.

The authors agree that there are multiple state-of-the-art high-throughput EM methods that could potentially lead to mm³ size datasets, however, to the best of our knowledge this is the first to achieve it. We have added more detailed comparison between several approaches in the *Introduction* section.

Specific comments & questions:

2. Abstract/ First sentence: 'Serial-section electron microscopy is the method of choice for studying cellular structure and network connectivity in the brain.' This exemplifies how biased the authors are about the technique, and dismisses 3-view block face method. They might have the opinion, but serial-section EM is a method... . Balance on other approaches is needed throughout the manuscript >6 occasions.

Reply: We modified the first sentence of the *Abstract* to "Electron microscopy is widely used for studying cellular structure and network connectivity in the brain... ." We have also made changes to balance on other approaches throughout the manuscript particularly in the *Introduction*.

3. Restructure the text to prevent duplications.

Reply: Following the reviewers advise we removed duplications throughout the manuscript.

4. Readability can be improved by fewer abbreviations and reducing numbers (e.g. make use of another table).

Reply: We have made several changes to the abbreviations. See also our reply to comment #10 bellow.

5. Automated acquisition after initial experimental setup (p4): how much time is needed for ‘initial experimental setup’?

Reply: We now address the reviewer question in the *Methods*: “After sectioning, the samples are loaded into the autoTEM, and the microscope pumps down to achieve desired vacuum. The pump down time depends on the number of sections being loaded onto the microscopes and for the large reels containing 5500 sections this process can take up to 24h. After vacuum is reached, we follow normal TEM operational routine to bring up the HT voltage and filament current and then align the beam. Calibration of the autoTEM includes camera pixels, stage and tape. After which, imaging can start. These calibration procedures cost roughly a day but are only required when changing the tape or filament.”

6. ‘TEMs achieve perhaps the highest signal-to-noise ratio, especially for fast imaging, but commercially available TEMs are neither designed nor optimized to efficiently image serial sections at a large scale.’ Compared to what?

Reply: Here we were comparing the commercially available TEMs with the piTEAM system. In the next sentence we argue that “Most commercial TEMs can hold just a few sections at one time and assume manual operation”, while with the system presented here can be loaded with 5500 sections and it is automated for non-stop continuous volume imaging with tens of thousands of tiles being acquired per montage. A task that would be impossible if done manually.

7. Page 5 is redundant and can be completely deleted.

Reply: Original Page 5 was substantially reduced and substituted by a light version of the summary at the end of the *Introduction*.

8. P6. Inches? Later feet? Please use metric system.

Reply: We now use the metric units.

9. Two stages mentioned: Voxa GridStage Sprite is not used to obtain data shown in this paper. Only the tape version is used. Why mention this?

Reply: GridStage Sprite was used in our pilot data collection (2500 sections of 250 μm \times 150 μm in size) on page 13: “The first dataset collected by piTEAM...”. We mention the GridStage Sprite

because some labs might want to use the speed improvements of our imaging platform without using tape methods. We have now added 2 citations for the dataset that has been collected with GridStage Sprite. This dataset is much smaller than the cubic millimeter volume, but it is larger than most available datasets, and has already given new insights into scientific discoveries which are not possible a few years ago. This setup using slot grids might be more applicable for many labs to conduct small volumes and answer questions of their own interests.

10. Many abbreviations make the text not easy to read (MSM TAO AWS CLAHE....) PiTEAM and pyTEM is confusing. Please use less uncommon abbreviations.

Reply: We have made several changes to the abbreviations.

We have eliminated the “AWS” (cloud service provider) and changed it to the more generic “cloud database”. We have reduced the occurrence of abbreviations and hope that improves the readability for audience. However, we would like to keep the following terms because they are novel and essential for our imaging pipeline. And here is a brief clarification of them:

- *piTEAM*: refers to the whole imaging pipeline, including both hardware (microscope, GridStage reel system) and software (operation system pyTEM, database, server etc.)
- *pyTEM*: operating system for imaging
- *MSM*: monitoring system created for the fleet of electron microscopes
- *TAO*: database for all the specimens and ROI's for the data collection

11. Another example of redundant text that can be deleted for clarity of the manuscript: P8. ‘In order to collect a petascale data set over months of continuous imaging, we designed the autoTEM system to be almost completely autonomous. In the sections below we describe the infrastructure that enabled this automation starting with the software.

Reply: We have removed this redundant text.

12. And another one: ‘In summary, the image acquisition software has been designed with the following goals in mind:

- **Completely autonomous operation. An autoTEM system should be able to run without user intervention beyond initializing the experiment.**
- **Adaptable to advances in sensor technology, enabling fast and affordable scaling. We have developed an abstracted software architecture, functional validation test plan, and change management plan to understand the impacts of sensor upgrades on image quality and effective throughput.**
- **Automatic ROI generation from optical images or low-mag EM images.**
- **Remote control of the TEM machines using web technologies.**
- **A cloud-based database, associating specimens, tapes, apertures, and ROIs which span optical and EM imaging modalities. This database manages the millions of**

images that are generated and tracks them back to ROIs, section ID, microscope configuration, imaging condition, etc.

- **Remote multi-system monitoring (MSM) tool that analyzes system and environmental variables (such as temperature and vacuum, among others) and uses feedback mechanisms.**

Reply: We removed the goals and kept key innovations in Box 1 on page 8-9.

13. I did not go into all detail in all the suppl. Data. Many of which seems redundant, exemplified by Fig. S4.

Reply: We have trimmed both *Methods* and *Supplemental Information* to remove redundancy and be more concise.

14. P14 Figure 4b should show deviation of expected tile position by colors. Very poorly visible. Please enlarge figure 4b.

Reply: Following the reviewer suggestion, we have modified the figure 4 (#3 in the revised manuscript)

15. P15. Fig 4e shows a fairly uniform focus map with tissue structures such as blood vessels being highlighted. This suggests that less homogeneous tissues (like pancreas, intestine) autofocus might fail. Also, additional satellite points (p 41) will not solve this. This will limit this approach to ‘homogeneous’ tissues like brain tissue, whereas the authors claim it is also suitable for other tissues (on p21).

Reply: Different tissue structures have different spatial frequencies. The autofocus can be fine-tuned by setting the min/max frequency range as well as the size of analysis area for Fast Fourier transformation. We currently crop center 2048×2048 pixels (comparing to 5408×5408 tile FOV) for autofocus, and choose the parameters for our images to achieve the best results. For less homogeneous tissues, one can crop larger areas to ensure the inclusion of features for analysis and adjust the frequency range for FFT calculation. The current tile FOV is $25 \mu\text{m} \times 25 \mu\text{m}$, which would be large enough for such studies. The satellite points of autofocus can also drop the locations with FFT score outliers during averaging. This autofocus algorithm has been working well so far for the brain tissues. Another strategy is to dynamically shift autofocus location at various zones to ensure some features are included, which is a quick change in the search algorithm.

16. Also the need of matching features (p15) for stitching limits this approach to a limited set of tissues. Tissues with (relatively) large areas which are poor in features (like lung, thyroid gland etc.) will fail in this workflow.

Reply: The real-time QC module described in this manuscript uses template matching instead of feature matching, which is more robust to areas poor in features. In fact, on-scope 2D stitching is

able to align on blood vessel through underlying textures of the support film. Moreover, the software lets you set failure thresholds to allow exclusively downstream QC and image processing. Though this downstream image processing is not the focus of this manuscript, even in samples which are poor in features, additional automated processes (filters, modified feature derivation parameters, even different algorithms to derive correspondence points) allows for match finding even on lower-contrast tissue.

It is unclear to the authors why the reviewer thinks the technique described in this manuscript would be more sensitive to tissue samples with low number of features, compared to any other competitive electron microscopy approaches, but we think we have implemented state of the art strategies to address such concerns.

17. P18. Upgradeability: Increasing amount of Pixels of CMOS sensors will at some point not necessary result in higher resolution images, because the details in the image on the phosphorescent screen will be limited. Especially for other approaches where ultrastructural details are important in contrast to brain connectomics where only heavily stained membranes are of interest. With the current setup in dataset very little details can be distinguished other than membranes as can be seen in the blow up in fig 5

Reply: For a fixed spatial resolution an increase in sensor size improves the imaging rating nearly linearly to a limit. This requires decreasing the imaging magnification of the electron microscope as well as a change in the fixed position of the camera optics to match the targeted pixel resolution. Certainly, there will be a limitation to the quality of the resolvable image features due to a combination of the grain size of the phosphor screen as well as the limitations of the electron optics at very low EM magnifications, but that limit is still far from being reached and there is a long life expectancy for this technology.

There are a few ways to try to increase the field of view at a fixed spatial resolution. First would involve better optimized electron optics on the scale of 800-1500x along with custom microscope apertures. Second, a smaller phosphor grain size can improve feature fidelity but decreases the photon count on the sensor thereby increasing the exposure time which might prove detrimental. Finally, a method to virtually expand the field of view can be accomplished by combining a grid multi camera images into one large field of view by raster scanning the electron beam without the speed penalty of discrete stage movements.

Regarding the reviewer comment on the details of the image, we have added to Figure 4 zoom in panels that show synapses and sub-cellular detail. We have also added *Supplementary Movie 4* that shows image tiles from pia to layer 6 at full resolution.

18. P19. Authors state their approach is superior to Multibeam SEM blockface imaging because their burst rate is a bit higher. However they do not mention that in a SEM with a slice and view approach the z resolution is twice as high. Also the stability of a FEG in a multibeam system is more stable compared to a LaB6 in a TEM and also has an approx. 20

times longer lifespan (~20 months vs 1 month) which is of high relevance if acquisition times are in the range of multiple months.

Reply: Each method has its own advantages and disadvantages. Speed of imaging has been widely advertised has the main advantage of the Multibeam SEM (blockface or serial section) and therefore we thought of using the same metric. In this revised version of the manuscript we added a more comprehensive comparison with other EM imaging methodologies in the *Introduction*.

Regarding the reviewer comment on filament stability, most of the downtime in those imaging approaches is not gauged by filament. The filament replacement in the systems described here takes up to 12 hours, which is very small comparing to 1~2 months of imaging time. We have also addressed this question on reviewer 1 comment #10.

19. P19. Please use metric units (15 feet)

Reply: We now use metric units

20. An issue not mentioned by the authors is image analysis. How to analyze petabytes of images? An additional algorithm to extract the relevant data and discard non relevant data should be very relevant. So on the fly building the connectomes for example.

Reply: On the fly building of connectomes is certainly an aspirational target. Though this is not possible yet, we now cite two manuscripts (Dorkenwald et al, 2019 and Schneider-Mizell, 2020) that show high quality dense segmentation of data generated with this pipeline. There are already automated tools developed for synapse detection. We are also in the process of building tools for auto-detecting neurons and classifying spines using machine learning.

21. If QC fails over and over because something went wrong with the section, how will this be dealt with?

Reply: The software will automatically mark the section as “QC failure” and abort the acquisition if QC fails repeated. The threshold of failures is a parameter that can be preset in the configuration file. Currently we are using 3 consecutive QC failures to prevent cascading section damage. This setting is empirical and based on engineering judgement.

22. New results on the XIMEA are resented in the discussion. Please restructure, put these into the results.

Reply: The focus of this manuscript is to present the completed volume montage datasets and the technical developments that enable such automated acquisition. Although the result on the new XIMEA camera is very exciting, the authors would like to keep it in the discussion due to the following reasons:

- New performance records are achieved quickly as we make tremendous efforts in engineering. We have made new breakthrough even when manuscript was under review during last few months, and expanded the ability to image non-square frame (another 20% increase comparing to what has been report in the manuscript). We think it is more important to present the concept of great scaling potential for the optics approach we adopted in our imaging pipeline.
- We are in the process of generating new datasets and would like to use the new AMT setup for collecting large volume serial montages. It would be more important for us to have the comparison of production level data acquisition performance instead of showing just a few beautiful images. The authors would be very excited to present those new results in the near future.

REVIEWERS' COMMENTS:

Reviewer #1 (Remarks to the Author):

The authors have undertaken substantial effort to address the reviewers' suggestions and their manuscript now appears in a much more consistent and comprehensible style.

The responses to the questions raised are satisfying. Considering some of the answers of general interest to the community, I urge the authors to include parts of their responses, in particular those covering the overall success/loss rate (questions 3, 5, 10) and the manpower needed to install and maintain the systems (q. 2) in their manuscript (discussion or methods section).

I can support publication, however suggest the authors to consider the following minor remarks:

Fig. 2 The text labels inside the figure panels are barely readable. Given the final panel size in print, they will appear as no more than a blurry streak of single pixels. Consider removing text and replacing it by symbols explained in the caption or re-arranging the figure. The journal's production team could support you with that.

Fig. 4f – please use the identical x-axis scaling and range for the two graphs.

Fig. 4g appears out of context. I suggest moving it to the supplement

Methods

Camera calibration: The pixel size calibrations are never exactly identical between different microscopes. How do you deal with these different xy pixel sizes when combining the volumes from different TEMs?

Aperture centroid finding: Does this happen in lowmag or at the acquisition magnification?

Reviewer #2 (Remarks to the Author):

The manuscript did significantly improve, but some points remain

- Correct for grammar
- Abstract: Conclusion? Future?
- For reasons of clarity SBEM should be SB-SEM when it is in the same sentence with FIB-SEM
- Intro has been improved significantly, but still not in balance regarding pro's and con's of other methods
- Intro is still too long (the final 1 page is summary/ results and should be deleted)
- Where can one access the data? What is the point of generating data if it is not accessible?

Dear Editors

Re: MS number NCOMMS-19-39169 "A Petascale Automated Imaging Pipeline for Mapping Neuronal Circuits with High-throughput Transmission Electron Microscopy "

We are submitting a revision of the above manuscript for consideration for publication in Nature Communications.

We have also incorporated most of the reviewers' suggestions and have made changes to the text in order to improve clarity in both the technological improvements and the associated details.

Below we address specifically all the reviewers' comments.

Reviewer #1

1. The authors have undertaken substantial effort to address the reviewers' suggestions and their manuscript now appears in a much more consistent and comprehensible style. The responses to the questions raised are satisfying. Considering some of the answers of general interest to the community, I urge the authors to include parts of their responses, in particular those covering the overall success/loss rate (questions 3, 5, 10) and the manpower needed to install and maintain the systems (q. 2) in their manuscript (discussion or methods section).

Reply: We have incorporated all these responses either on the main text or in the supplementary information.

2. Fig. 2 The text labels inside the figure panels are barely readable. Given the final panel size in print, they will appear as no more than a blurry streak of single pixels. Consider removing text and replacing it by symbols explained in the caption or re-arranging the figure. The journal's production team could support you with that.

Reply: We have increased the fonts in figure 2.

3. Fig. 4f – please use the identical x-axis scaling and range for the two graphs.

Reply: We have made the figure changes suggested by the reviewer.

4. Fig. 4g appears out of context. I suggest moving it to the supplement.

Reply: We would like to keep figure 4g in the main document as the fast step and settle of our system is an advantage in speed improvement compared with other alternatives.

5. Camera calibration: The pixel size calibrations are never exactly identical between different microscopes. How do you deal with these different xy pixel sizes when combining the volumes from different TEMs?

Reply: Microscope calibration and auto-focus leads to a pixel size variation across microscopes of up to 0.4 nm. Although inter-section alignment is not in the scope of this manuscript, our alignment strategy effectively normalizes the pixel size by implementing a series of regularized transformation solutions which strike a balance minimizing image deformation and residuals of point correspondences derived from scale-invariant descriptors. It is possible for us to use alignment metadata to map pixel size changes from raw data to this remapped volume space.

6. Aperture centroid finding: Does this happen in lowmag or at the acquisition magnification?

Reply: The aperture centroid finding is performed at the montage acquisition magnification. We have added this to the “Aperture Centroid Finding” under Methods section.

Reviewer #2

1. Correct for grammar

Reply: The manuscript was checked for grammar mistakes.

2. Abstract: Conclusion? Future?

Reply: We have revised the abstract to more clearly mention our conclusion and future directions

3. For reasons of clarity SBEM should be SB-SEM when it is in the same sentence with FIB-SEM.

Reply: SBEM was replaced with SB-SEM.

4. Intro has been improved significantly, but still not in balance regarding pro's and con's of other methods.

Reply: We think the section is well balanced but added a further revision.

5. Intro is still too long (the final 1 page is summary/ results and should be deleted)

Reply: It is our understanding that the paragraph of Introduction needs a brief summary of both the results and the conclusions, which is a requirement from Nature Communications. The authors have trimmed the content to make it more concise.

6. Where can one access the data? What is the point of generating data if it is not accessible?

Reply: We describe this in the data availability section.